# Hyperbranched Polymer Network Based on Electrostatic Interaction for Anodes in Lithium-Ion Batteries

**DOI:** 10.3390/ma15227921

**Published:** 2022-11-09

**Authors:** Chenchen Yang, Yan Jiang, Na Cheng, Jianwei Zhao, Feng Chen

**Affiliations:** 1School of Materials Science and Engineering, Changzhou University, Changzhou 213164, China; 2School of Materials and Textile Engineering, Jiaxing University, Jiaxing 314041, China

**Keywords:** hyperbranched polyethyleneimine, polymer binder, pH value, electrostatic interaction

## Abstract

Silicon is considered as one of the ideal anode materials for the new generation of lithium-ion batteries due to its extremely high theoretical specific capacity. Nevertheless, in the actual charging and discharging process, the Si electrode will lose its electrochemical performance due to the huge volume change of Si nanoparticles resulting in detachment from the surface of the fluid collector. The polymer binder can bond the Si nanoparticles together in a three-dimensional cross-linking network, which can thus effectively prevent the Si nanoparticles from falling off the surface of the fluid collector due to the drastic change of volume during the charging and discharging process. Therefore, this study developed a new polymer binder based on electrostatic interaction with hyperbranched polyethylenimine (HPEI) as the main body and water-soluble carboxylated polyethylene glycol (CPEG) as the cross-linker, where the degree of cross-linking can be easily optimized by adjusting the pH value. The results demonstrate that, when the density of positive and negative charges in the binder is relatively balanced at pH 7, the stability of the battery’s charge–discharge cycle is significantly improved. After 200 cycles of constant current charge–discharge test, the specific capacity retention rate is 63.3%.

## 1. Introduction

Faced with the increasingly serious environmental problems, including the greenhouse effect, many countries attach great importance to it. Among them, the widespread use of conventional fuel vehicles is one of the main reasons [1,2,3]. Therefore, the research and development of electric vehicles is of particular importance. With the development of science and technology, batteries have been applied in many different fields, such as new energy vehicles, biosensors, and flexible wearable devices [4]. It is of note that lithium-ion batteries with higher energy density exert a key role in this field. This has proposed high requirements for anode materials [5]. As a candidate anode material, Si is considered as the most promising anode material due to its high theoretical specific capacity (4200 mAh g^−1^) [6]. Nevertheless, the volume of Si varies greatly (>400%) during the charging and discharging process, making it difficult to apply in practice at present [7,8].

Therefore, numerous studies have aimed to develop a highly efficient polymer binder that can withstand drastic volume changes of Si nanoparticles and prevent the analytical shedding of Si electrodes. Compared with traditional linear polymers including sodium carboxymethyl cellulose (CMC) [9], polyvinyl alcohol (PVA) [10], polyacrylic acid (PAA) [11,12,13,14], chitosan [15] and polysaccharide [16], hyperbranched polymers can form stronger interaction with Si nanoparticles due to their 3D structure and high functional group density [17], thus exhibiting more obvious advantages [18].

Recently, a new polymer binder based on hyperbranched polyethylene imine (HPEI) has been proposed [19,20]. In the current work, the chemical crosslinking of HPEI was employed to form a 3D network structure [21]. The hyperbranched structure of HPEI gives it a higher functional group density, which enables HPEI to generate stronger hydrogen bonding with silicon particles compared with traditional linear polymers. In the meantime, the rich functional group density also makes higher adhesion strength between polymer binder and copper foil. Recent research shows that the pH value of the polymer binder plays an extremely important role in adapting to the large volume changes of the Si electrode during cyclic charging and discharging [22,23,24]. However, this finding has been previously ignored. In this study, when the pH value is too high, the protonation degree of the amino group in HPEI is too low, and the positive charge density in the cross-linking network is extremely low. At this time, the deprotonation degree of the carboxyl group in carboxylated polyethylene glycol (CPEG) is high, and the negative charge density is high. When the pH is too low, the opposite is true, and the density of positive charges is much higher when compared with the density of negative charges. Obviously, these two conditions are not conducive to the formation of stable three-dimensional network structure through electrostatic interaction of positive and negative charges. Therefore, it is necessary to adjust the pH value within an appropriate range, so that the positive and negative charge densities are relatively balanced. This facilitates to achieve the best cross-linking degree, and subsequently the maximum extent of silicon powder and carbon black of the package, finally improving the performance of the battery. As shown in Figure 1.

In this paper, hyperbranched polyethyleneimine (HPEI) was used as the main body, and carboxylated polyethylene glycol (CPEG) was used as the cross-linking agent. By simultaneously optimizing the pH value and adjusting the density of positive and negative charges, the hyperbranched three-dimensional cross-linking network prepared finally significantly improved the battery performance. Under the current density of 500 mA g^−1^, the charging capacity of the silicon negative electrode can still reach 838 mAh g^−1^ after 200 cycles, which has good cyclic charging and discharging stability.

## 2. Experiments

Silicon powder particles (average particle size of ~100 nm) were purchased from Beijing Deke Daojin Science and Technology Co., Ltd. (Beijing, China). Hyperbranched Poly(ethyleneimine) solution (HPEI, number average molecular weight Mn ~60,000 by GPC, 50 wt% in H_2_O) was obtained from Sigma-Aldrich (St. Louis, MO, USA). The cross-linkers Carboxylated polyethylene glycol was obtained from Guangzhou Tanshui Technology Co., Ltd. All materials were used as received without further purification (Guangzhou, China).

### 2.1. Preparation of Hyperbranched Gel

Firstly, 2 g HPEI aqueous solution was dissolved in 8 mL deionized water in order to obtain 0.1 g/mL HPEI aqueous solution. Then, 20 wt% (relative to HPEI) of crosslinking agent CPEG was added to the solution, and the solution state was stirred at room temperature until it remained constant, and the change of state was recorded. Finally, the pH values were adjusted to 6.0, 7.0, 8.0, and 10.0 by using the 12% hydrochloric acid solution, respectively.

### 2.2. Fabrication of Si Electrode

Taking pH 6.0 as an example, the mixing process was carried out manually, grinding a mixture of cross-linking solution, Si nanoparticles, and acetylene black at a mass ratio of 60:20:20 in an agate mortar for 30 min to obtain a uniform slurry for coating. Subsequently, the resulting slurry was coated on copper foil and dried for 24 h at room temperature. Finally, the silicon negative electrode was obtained after drying within a vacuum oven at 70 °C for 12 h. The process of silicon anode preparation for other pH values is similar.

### 2.3. Material Characterization

Attenuated total reflection Fourier transform infrared (ATR-FTIR) characterization was performed using the FTIR spectrometer. Moreover, field emission scanning electron microscopy (FE-SEM, Hitachi, Tokyo, Japan, S-4800) was conducted to observe the surface morphology of silicon negative electrodes at different pH values before and after cyclic charging and discharging. Thereafter, the tensile mechanical properties of the cross-linked dry films were tested on a general material tensile machine (Suns, 2503) at a strain rate of 20 mm min^−1^ under ambient atmospheric conditions.

### 2.4. Peel Test

To determine the adhesion of each adhesive, a 180° peel test was performed using a universal stretching machine (UTM). The silicon negative electrode was first cut into 10 × 50 mm test strips. Then, 3M tape (10 mm width) was adhered to the laminate of the electrode. Finally, the tape was pulled at a peeling rate of 20 mm min^−1^ to record the required strength in order to separate the laminate layer from the copper foil.

### 2.5. Electrochemical Characterization

The electrochemical performance was evaluated and the fabricated silicon negative electrode was assembled into a 2025 button cell in an argon-filled glove box. Here, the silicon negative electrode was used as the working electrode, the lithium sheet as the reference electrode, and the 1 mol/L LiPF_6_ solution as the electrolyte, whereas the polypropylene film as the diaphragm. Additionally, a NEWARE V00749 test system was employed for a constant current charge and discharge cycle over a potential range of 0.01–2 V. After the cycle was formed, the cycle charging and discharging performances of the battery were evaluated at different current rates. In addition, electrochemical impedance spectroscopy (EIS) measurements were performed on a potentiostat (CHI, 660E) at the frequencies of 0.1 Hz to 100 kHz.

## 3. Results and Discussion

### 3.1. Synthesis and Characterization of the Elastic Cross-Linked Binder

The crosslinking reaction between HPEI and CPEG was confirmed by infrared spectrum analysis. The spectra of CPEG and HPEI and HPEI-CPEG (Figure 1) exhibit the broad absorption bands of the COOH and/or NH stretching from 3600 to 3100 cm^−1^, the CH stretching from 3000 to 2860 cm^−1^, the broad N-H bending from 1683 to 1604 cm^−1^, and the CH bending from 1182 to 1003 cm^−1^; meanwhile, the N-H peak intensity of HPEI-CPEG was lower than that of HPEI. In addition, the decrease of peak intensity indicated that the amino group in HPEI had successfully reacted with the carboxyl group in CPEG [25,26].

HPEI-CPEG-pH6, HPEI-CPEG-pH7, HPEI-CPEG-pH8, and HPEI-CPEG-pH10 were characterized by FE-SEM. The effect of crosslinking on morphology was directly observed. As shown in Figure 2a, the size of Si nanoparticles is around 160 nm. In the images of HPEI-CPEG-pH6, HPEI-CPEG-pH8, and HPEI-CPEG-pH10 (Figure 2a,c,d), Si nanoparticles produced a large number of agglutinations, and the surface was covered with a thick organic layer. Meanwhile, silicon powder particles had an agglomeration phenomenon and uneven distribution. The Si nanoparticles could not be effectively dispersed in the matrix. However, as observed from the HPEI-CPEG-pH7 image (Figure 2b), Si nanoparticles were uniformly dispersed in the matrix, indicating that intact 3D cross-linking networks were formed in the cross-linking process to effectively prevent the aggregation of low-molecular weight hyperbranched PEI. The HPEI-CPEG-pH7 coating can also buffer the volume changes of silicon powder particles and limit their movement to maintain the structural integrity of the silicon negative electrode [27].

Adhesion is another important performance index of the polymer binder. Here, the HPEI-CPEG-pH6, HPEI-CPEG-pH7, HPEI-CPEG-pH8, and HPEI-CPEG-pH10 silicon negative electrode were prepared, and then a 180° peeling test was performed to evaluate the adhesion performance. Results in Figure 3 suggested the high pH generated a low protonation degree of the amino group in HPEI; as a result, a low positive charge density was created in the HPEI; in the meantime, there was a sufficient carboxyl group in the polyethylene glycol (PEG) deprotonated to cause high negative charge density, eventually making too weak electrostatic interaction to effectively form the three-dimensional cross-linked network [28]. Consequently, when the pH value was higher than 8.0, the difference between positive and negative charge densities gradually increased with the increase in pH value, and the strength of electrostatic cross-linking gradually decreased, leading to the direct mechanical stress on the HPEI and CPEG backbone. Therefore, the polymer chain is not difficult to fall off from the copper foil. When the pH value was below 7.0, due to the further reduction in pH value and the continuous increase in protonation degree of amino group along with a decreasing deprotonation degree of the carboxyl group, the difference between positive and negative charge density was constantly expanding, which weakened the role of electrostatic action, thus affecting the entire three-dimensional cross-linked network structure. The adhesive force of HPEI-CPEG-pH7 reached the maximum value of 1.102 N/mm^2^. The excellent bonding performance of HPEI-CPEG-pH7 shows that the density of positive and negative charges in the three-dimensional cross-linking network of the binder is relatively balanced at the pH value of 7.0, ensuring the strength of electrostatic cross-linking in the whole network. In this case, the fully cross-linked binder network can evenly distribute the mechanical stress to a large number of anchor points on the branch chain, thus significantly reducing the stress imposed on each anchor point, and finally making the adhesive properties of the binder significantly improved.

The sliding degree of the branch chain of HPEI after cross-linking exerts a vital role in maintaining the mechanical stability of the silicon negative electrode. Here, a universal stretching machine (UTM) was employed to investigate the influence of pH value on the mechanical properties of the HPEI-CPEG three-dimensional cross-linking network. The tensile properties of HPEI-CPEG-pH6, HPEI-CPEG-pH7, HPEI-CPEG-pH8, and HPEI-CPEG-pH10 were obtained (Figure 4). According to the figure, with the gradual optimization of pH value, the tensile strength of the binder film also improved. When the pH is 7.0, the compression modulus increases around 1.5 times. This is mainly because, under this pH condition, the densities of positive and negative charges are relatively balanced, and the electrostatic cross-linking node of hyperbranched 3D cross-linking network reaches the peak, which effectively limits the sliding of the HPEI chain, finally significantly improving the mechanical stability of the silicon negative electrode.

### 3.2. Electrochemical Properties

To investigate the electrochemical performance of HPEI-CPEG as a silicon negative electrode binder, several constant–current charge–discharge tests were carried out in the voltage range of 0.01–2.0 V. In these tests, the HPEI-CPEG cross-linking network at different pH values was selected as the silicon negative electrode binder. The initial constant current curve is shown in Figure 5. All the samples exhibited the characteristic silicon plateau at a voltage of approximately 0.1/0.5 V. Initial Coulombic efficiency of HPEI-CPEG-pH6, HPEI-CPEG-pH7, HPEI-CPEG-pH8, and HPEI-CPEG-pH10 ICE were 75.8%, 93.7%, 86.6%, and 80.6%, respectively. As revealed by the HPEI-CPEG-pH6 minimum value of the ICE, the silicon surface consumed a large amount of electrolyte during the first charge and discharge cycle. This is possibly because, when the pH is too low, the high positive charge density in HPEI and low negative charge density in CPEG lead to low connections in the entire cross-linked network, and the surface of silicon powder particles is not well covered. Compared with other electrodes, the HPEI-CPEG-pH7 sample has a better ICE value, indicating that the positive charge density of the protonated amino group in HPEI is relatively balanced with the negative charge density of the deprotonated carboxyl group in carboxyl PEG under appropriate pH conditions. At this time, the cross-linked network formed can promote the generation of multi-dimensional hydrogen bonding interaction between a high-density amino group of HPEI and the surface of Si nanoparticles via the sufficient electrostatic cross-linking nodes, thus reducing the decomposition caused by the contact between silicon and electrolyte. In addition, the hyperbranched network structure also limits the movement of Si and lowers the tendency of Si nanoparticles to fall off the collector surface [29,30].

To compare the charge–discharge cycle stability of silicon negative electrodes, the specific capacity retention rates of different silicon negative electrodes after several charge–discharge cycles were studied by a cycle test at the current density of 500 mA g^−1^. As shown in Figure 6, in the initial activation stage (the first 10 cycles), the Si volume changed greatly [30], the solid electrolyte interface layer (SEI) was formed, the electrode structure was changed, and the capacity of all samples decreased significantly. The specific capacity of the HPEI-CPEG-pH10 electrode decreased to 691 mAh g^−1^ after only 10 cycles. To improve the mechanical stability of the HPEI-CPEG 3D cross-linked network, an appropriate pH value should be determined, so that the cross-linking density of the binder reached the peak value. However, the negative cycle stability of HPEI-CPEG-pH10 silicon was poor at the pH value of 10.0, which indicated that the difference between positive and negative charge densities in the binder was too large, resulting in the too low cross-linking density, thus making it impossible to form an effective cross-linked network structure. When the pH value is controlled at 6.0, the specific capacity of the HPEI-CPEG-pH6 silicon negative electrode decreases to 531 mAh g^−1^ after 200 cycles. The poor cyclic stability of the silicon negative electrode can also be ascribed to the low cross-linking density caused by the imbalance between positive and negative charge densities in the binder due to the inappropriate pH value. Finally, due to the high cross-linking density generated by the relative balance of positive and negative charge densities, the silicon negative electrode with pH 7.0 shows the best charge–discharge cycle stability, with a specific capacity of 838 mAh g^−1^ after 200 cycles. The hyperbranched structure of HPEI makes the amino group form multi-dimensional hydrogen bonds with Si. Additionally, HPEI-CPEG-pH7 forms a complete three-dimensional cross-linking network, which effectively inhibits the displacement of Si nanoparticles and makes the deformation of Si negative electrode smaller when the volume changes. Therefore, at the pH value of 7.0, the resulting cross-linked network not only improves the mechanical stability of HPEI, but also maintains its hydrogen bond reversibility.

To evaluate the magnification performance of the silicon negative electrode, the specific capacity of the silicon negative electrode was also investigated at different pH values at current densities from 500 to 2000 mA g^−1^ (Figure 7). Firstly, the specific capacities of HPEI-CPEG-pH6, HPEI-CPEG-pH7, HPEI-CPEG-pH8, and HPEI-CPEG-pH10 decreased gradually with the increase of current density. After further comparison, it can be observed that, when the current density reaches 2000 mA g^−1^, the silicon negative electrode of HPEI-CPEG-pH6, HPEI-CPEG-pH7, HPEI-CPEG-pH8, and HPEI-CPEG-pH10 retain the specific capacity of 589, 890, 383, and 147 mAh g^−1^, respectively. When the current density is restored to 200 mA g^−1^, the specific capacities of HPEI-CPEG-pH6, HPEI-CPEG-pH7, HPEI-CPEG-pH8, and HPEI-CPEG-pH10 are restored to 773, 1028, 576, and 317 mAh g^−1^, respectively. In conclusion, the excellent rate performance of HPEI-CPEG-pH7 may be attributed to the high-density three-dimensional cross-linked network and the high lithium-ion conductivity of HPEI. Notably, the HPEI-CPEG-pH7 of high-density three-dimensional cross-linked network not only prevents silicon powder particles from crushing and cracking in the negative electrode, but also promotes the rapid electrolyte diffusion to the surface of silicon powder particles [31].

To further determine the influence of binder composition and cross-linking density on recharging performance, EIS was employed to test the diffusion kinetics of Si/HPEI and Si/cHPEI negative terminals. Figure 8 displays the Nyquist plot after the 200th discharge. Both curves in the figure contain a high frequency semicircle (HFS) and an intermediate frequency semicircle (MHS), which overlap each other to form a semicircle, and a long low frequency straight line (LFL). The above-mentioned HFS, MHS, and LFL correspond to SEI resistance, charge transfer resistance, and Warburg impedance of lithium-ion diffusion in solid materials, respectively. The equivalent circuit is displayed in the figure, where R_s_, R_SEI_, R_ct_, and W represent Warburg impedance of electrolyte resistance, SEI resistance, charge transfer resistance, and solid phase diffusion, respectively. Through equivalent circuit fitting of the impedance map, the diffusion coefficient D of lithium ion in the negative electrode can be obtained using Formula (1). D reflects the difficulty in lithium-ion diffusion in the negative electrode, and the stability of binder and conductive three-dimensional network structure in the charging and discharging process [32,33]:D = R^2^T^2^/2A^2^n^2^F^4^C^2^δ^2^(1)
where R, T, and F are gas constant, absolute temperature, and Faraday constant, respectively; A is electrode surface area; C is the molar concentration of lithium ions; Using the real part of Warburg impedance against the negative square root of the angular frequency to plot the linear relationship, Warburg coefficient δ can be obtained according to the slope of the line [34].

After calculation, it can be found that, after cycling, the D values of HPEI-CPEG-pH6, HPEI-CPEG-pH7, HPEI-CPEG-pH8, and HPEI-CPEG-pH10 are shown in Table 1. The reduced D value may be caused by the crushing and agglomeration of silicon powder particles, which can hinder the diffusion of lithium ions in the negative electrode. Among all the samples, the HPEI-CPEG-pH7 electrode has the highest D value, suggesting that the hyperbranched three-dimensional cross-linking network of HPEI-CPEG-pH7 has the most stable structure during the charging and discharging cycle, which ensures the rapid transport of lithium ions in the silicon negative electrode, improving the charging performance of the silicon negative electrode [9].

To more directly determine the interfacial stability of silicon negative electrode, the surface morphology of Si anode using HPEI-CPEG-pH6 (Figure 2e), HPEI-CPEG-pH7 (Figure 2f), HPEI-CPEG-pH8 (Figure 2g), and HPEI-CPEG-pH10 (Figure 2h) for 100 cycles was observed by FE-SEM. After 100 cycles of charging and discharging, the surfaces of all samples were covered with a thick SEI film. In contrast, a large number of Si nanoparticles were still clearly observed on the surface of HPEI-CPEG-pH7 (Figure 2f). Observations of these surface morphologies support the idea that the cross-linked network can effectively buffer the dramatic volume changes in Si powder particles and limit the movement of Si powder particles at the optimized pH values, thus maintaining the micro topography of the negative electrode and greatly improving the stability of the silicon negative electrode during the charging and discharging cycle [7].

## 4. Conclusions

In this study, a hyperbranched polymer with a 3D cross-linked network was successfully screened as an efficient binder for the Si negative electrode in lithium-ion batteries, and its pH effect was investigated. It was found that the pH value played a crucial role in the formation of the hyperbranched polymer three-dimensional cross-linking network. The optimized cross-linking network significantly reduced the agglomeration tendency of HPEI and formed a 3D network with high enough cross-linking density, which effectively buffered and inhibited the volume change of Si powder particles. In this cross-linking network, the mechanical stress can be evenly distributed to each branch chain of HPEI, which can greatly improve the buffer effect of the binder on the volume expansion of Si. As a result, after optimization, we adjusted the pH value in an appropriate range, and thus the positive charge density of protonated amino group in HPEI and the negative charge density of deprotonated carboxyl group in CPEG are relatively balanced to form a stable hyperbranched three-dimensional cross-linked network structure through electrostatic interaction. Finally, silicon powder particles and carbon black are wrapped to the maximum extent, and the stable performance of battery charging and discharging cycle is improved. Moreover, this provides a new perspective in the field of silicon carbon batteries in the future.

## Data Availability

The data presented in this study are available on request from the corresponding author.

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
