# Peer review of "Hyperbranched Polymer Network Based on Electrostatic Interaction for Anodes in Lithium-Ion Batteries"

_materials, 2022, doi:10.3390/ma15227921_

Round 1
Reviewer 1 Report
This manuscript reports on the impact of application of hyperbranched polymer network based on electrostatic interaction for anodes in lithium-ion batteries. The topic is interesting and the conclusions seem sound. However, the manuscript lacks clarity in its present form. Therefore, the following issues should be addressed before publication.
1 In my opinion, in the introduction, the author should discuss in more detail the reasons for choice of the starting compounds as well as their advantages compared to analogues employed by other research teams. Perhaps, it would be better to schematically present the subject of research, namely, the anode, and to show its part that the authors aim to investigate.
2 In the experimental part, it is indicated that agate mortar was used to prepare uniform slurry of cross-linking solution with Si nanoparticles and acetylene black. Was the mixing process itself carried out manually or with the help of a special mill, whereto the mortar was placed?
3 Figure 1 shows the IR spectra. The authors should provide a more detail assignment of each registered signal, for example, in the table. For clearer understanding, it would be better to give the values of wavelengths next to the absorption bands.
4 Is it possible to attribute a 160 nm particle to nanoparticles? After all, according to the generally accepted definition, a nanoparticle is usually defined as a particle of matter that is between 1 and 100 nanometres (nm) in diameter [https://en.wikipedia.org/wiki/Nanoparticle#cite_note-epaXXXXa-1].
5 For better understanding of the described materials, it would be very useful to give a scheme showing distribution of the nanoparticles in layers of the hyperbranched polymer network, distribution of the layers themselves, and implementation of these process in the created anode.
6 The paper is not free from annoying typos, which need to be corrected. For example, symbols that should be in lowercase or uppercase are given as lower case letters, namely "H2O" on line 70, or "g-1" on line 222, etc. The text contains words in the national language, for example, "和" in line 163, etc. Some references miss "[]", for example, in line 144. In the captions for Figures 3-7 and Table 1, extra spaces are placed in the names of the samples, which makes it difficult to read.
7 The information shown in Figures 2 and 9 should be presented in a convenient way for comparison. It may be better to combine in these figures into one.
Author Response
Dear reviewer:
Thank you very much for your fruitful suggestions on our manuscript materials-2033855. According to the suggestions, we have revised the manuscript carefully. Please see the attachment.
Yours sincerely,
Pro. Jianwei Zhao

Reviewer 2 Report
Comments:
1. Insert a Table showing the comparison of ionic conductivity, ion transference number, and voltage window.
2. Cite the following references in the introduction section
https://doi.org/10.1016/j.mset.2018.08.001
https://doi.org/10.1016/j.jpba.2022.115120
3. Insert a scheme showing the polymer binder bonding the Si nanoparticles in a three-dimensional cross-linking network.
4. Explain the nature of Fig. 5 obtained at different pH for HPEI-CPEG.
5. Write a section on the electrochemical properties to investigate the performance and the influence of cross-linking.
Author Response

(The authors gave the same response as above.)
